# Hyperflux:
# Pruning Reveals the Importance of Weights

## Abstract

Network pruning is used to reduce inference latency and power consumption in large neural networks. However, most existing methods use ad-hoc heuristics, lacking much insight and justified mainly by empirical results. We introduce Hyperflux, a conceptually-grounded $L_0$ pruning approach that estimates each weight's importance through its *flux*, the gradient's response to the weight's removal. A global *pressure* term continuously drives all weights toward pruning, with those critical for accuracy being automatically regrown based on their flux. We postulate several properties that naturally follow from our framework and experimentally validate each of them. One such property is the relationship between final sparsity and pressure, for which we derive a generalized scaling-law equation that is used to design our sparsity-controlling scheduler. Empirically, we demonstrate state-of-the-art results with ResNet-50 and VGG-19 on CIFAR-10 and CIFAR-100.

## 1   Introduction

Overparameterization has become the norm in modern deep learning to achieve state-of-the-art performance [35, 2, 25]. Despite clear benefits for training, this practice also increases computational and memory costs, complicating deployment on resource-constrained devices such as edge hardware, IoT platforms, and autonomous robots [42, 26]. Recent theoretical and empirical findings suggest that sparse subnetworks extracted from large dense models can match or exceed the accuracy of their dense counterparts [7, 58, 33, 24, 5, 4, 55, 8, 51] and even outperform smaller dense models of equal size [37, 27, 59]. These results have created interest in network pruning as a strategy to identify minimal, high-performing subnetworks.

Pruning has a rich history [22, 34, 47] and continues to prove valuable for real-time applications [13, 19, 50]. Recent methods have significantly advanced the field by resorting to a variety of strategies and heuristics, from magnitude pruning, gradient methods, and Hessian-based criteria [12, 13, 23, 43, 3, 7] to dynamic pruning approaches [29, 4, 40, 21, 52] or combinations thereof [30, 6]. However, the strong interdependence between weights remains a challenge [18, 46, 24, 5, 31], as it complicates the task of determining each weight's importance. Optimal pruning has been explored [16, 23], but such formulations are typically computationally intractable in practice. In contrast, most current state-of-the-art strategies prioritize empirical results and speed through heuristics, at the expense of theoretical grounding.

Given this gap, we ask: *Can we create a pruning method that is both empirically strong and conceptually grounded?*

Inspired by the principle that *the value of something is not truly known until it is lost*, which has shaped major discoveries in fields such as functional genomics [10, 41], neuroscience [38], and network science [1], we introduce Hyperflux, an $L_0$ pruning method that determines a weight's

importance by first removing it. Unlike most works, Hyperflux puts a large emphasis on conceptual grounding and explainability.

The *main idea* of our method is that each weight has a *flux*, which appears when the weight is pruned through the network's gradients. A global $L_0$ regularization term called *pressure* pushes all weights towards pruning, aiming to uncover each of their fluxes. Those weights whose flux is greater than the pressure will be regrown, while the rest will remain pruned. This process is repeated until the end of training. A useful side effect of pruning and regrowth happening concurrently on all weights multiple times is that the network's topology implicitly becomes noisy, disentangling the overall weight evaluation from a specific topology.

We postulate several properties that emerge from our framework: sparsity convergence, a sparsity-pressure relationship, and large flux for important weights. We empirically confirm each of these properties and, for the sparsity-pressure relationship, we obtain dependencies similar to those of known scaling laws in neural networks [15, 20, 48, 39, 11, 14, 56, 17]. Based on the postulated properties, we propose a pressure scheduler, as well as a stabilization stage after pruning, further differentiating Hyperflux from recent $L_0$ methods [32, 40, 54, 29]. The scheduler is used to achieve the desired sparsity, after which the stabilization stage recovers accuracy lost to noise induced by pruning.

Summarizing, our key contributions are:

- We introduce *Hyperflux*, a conceptually grounded pruning method which develops the notions of *flux* and *pressure*, before empirically studying their emergent properties.
- Based on these properties, we introduce a pressure-controlling scheduler to achieve a desired sparsity, as well as a stabilization stage after pruning.
- We obtain state-of-the-art results, achieving better or comparable accuracy to existing methods in empirical validation across several networks and datasets.

## 2  Related work

Research on neural network pruning has a relatively old history, with some methods going back decades and laying the groundwork for modern approaches. Early approaches, such as [22] and [23], utilized Hessian-based techniques and Taylor expansions to identify and remove unimportant specific weights, while [34] employed derivatives to remove whole units, an early form of structured pruning. These initial studies demonstrated the feasibility of reducing network complexity without significantly compromising performance. An influential overview [47] concluded that magnitude pruning was particularly effective, a paradigm that since then has been widely adopted [13, 7, 58, 6, 21, 12, 44, 36, 9].

**The existence of highly effective subnetworks** builds upon these foundational studies, with the Lottery Ticket Hypothesis [7] being a good example. This work uses magnitude pruning to demonstrate that there exists a mask which, if applied at the start of training, produces a sparse subnetwork capable of matching the performance of the original dense network after training, if the initialization is kept unmodified. Subsequent research has further validated this concept by showing that these subnetworks produced by masks, even without any training, achieve significantly higher accuracy than random chance [58], reaching up to 80% accuracy on MNIST. Moreover, training these masks instead of the actual weight values can result in performance comparable to the original network [37, 58], suggesting that neural network training can occur through mechanisms different from weight updates, including the masking of randomly initialized weights. Other studies have attempted to identify the most trainable subnetworks at initialization. SNIP [24] use gradient magnitudes as a way to identify trainable weights, while [40] employ $L_0$ regularization along with a sigmoid function that gradually transitions into a step function during training, enabling continuous sparsification. These findings indicate that the specific values and even the existence of certain weights may be less critical than previously believed.

**Dynamic pruning** differs from classical heuristics by allowing the model to make pruning decisions while processing the input, without a fixed pattern. Some methods use learnable parameters, e.g. [21] train magnitude thresholds for each layer in the network to determine which weights will be pruned. Other works, like that of [4], do not have any learnable parameters, learning instead a weight distribution whose shape will determine which and how many weights are pruned. Yet another class

of $L_0$ regularization techniques [40, 32] try to maximize the number of removed weights. Hyperflux aligns with the dynamic pruning paradigm by enabling continuous pruning of weights based on learnable parameters. However, unlike such methods, Hyperflux does not treat the regularization as a fixed value, but as an adjustable input of the training procedure, which can be used to control its behavior.

**Pruning based on gradient values** is another prominent approach, often overlapping with dynamic methods, which assesses weight properties in relation to the loss function. Works [24] and [5] assess the trainability of subnetworks by analyzing initial gradient magnitudes relative to the loss function. AutoPrune [53] introduces handcrafted gradients that influence training, while Dynamic Pruning with Feedback [28] uses gradients during backpropagation to recover pruned weights with high trainability, preserving accuracy. RigL [6] use gradient and weight magnitudes to determine which weights to prune and to regrow. GraNet [30] employs a neuroregeneration scheme, which prunes and regrows the same number of weights, effectively keeping the sparsity constant while growing accuracy. Hyperflux distinguishes itself from all these methods by evaluating the importance of weights *after* the moment of their pruning. Instead of deciding which weights are (un)important based solely on instantaneous gradients or single-stage evaluations, Hyperflux identifies a weight's significance based on the aggregated impact across topologies its removal has on the network's performance.

# 3 Hyperflux method

We associate each weight $\omega_i$ to a learnable parameter $t_i$, which determines whether the weight is present ($t_i > 0$) or pruned ($t_i \leq 0$). We define a weight's importance to be the increase in loss caused by its pruning. We assess the importance of a weight $\omega_i$ through its flux, the gradient of $t_i$ with respect to the loss function when $t_i \leq 0$. The connection between flux and weight importance is detailed in Section 3.2. The *pressure* term, denoted by $L_{-\infty}$, will push all $t$ values towards $-\infty$, pruning the weights and revealing their fluxes. No manual selection or analysis of gradients is needed, since the interaction between pressure and flux during backpropagation will naturally only keep important weights whose flux is large.

## 3.1 Preliminaries

Consider a neural network defined as a function $f : \mathcal{X} \times \mathbb{R}^d \to \mathcal{Y}$ where $\mathcal{X}$ is the input space, $\mathcal{Y}$ is the output space, and $\mathbb{R}^d$ is the space of weights. Given a training set $\{(x_j, y_j)\}_{j=1}^J$, learning the weights $\omega$ amounts to minimizing a loss function so that $f(x_j, \omega)$ aligns with $y_j$:

$$\mathcal{L}(\omega) = \sum_{j=1}^J \ell\big(f(x_j, \omega), y_j\big),$$

We define the topology of the neural network as a binary vector $\mathcal{T} \in \{0, 1\}^d$ where $\mathcal{T}_i$ represents whether weight $\omega_i$ is pruned or not. We denote a family of topologies as $\mathcal{T}^{1 \to K}$, with $K$ its cardinality and $\mathcal{T}^k$ a specific topology from the family. Thus, the loss of a network with topology $\mathcal{T}$ is:

$$\mathcal{L}(\omega, \mathcal{T}) = \sum_{j=1}^J \ell\big(f(x_j, \omega \odot \mathcal{T}), y_j\big),$$

where $\odot$ is the Hadamard product. For each weight $\omega_i$, we introduce a learnable presence parameter $t_i$ with $t \in \mathbb{R}^d$ denoting the vector collecting all $t_i$. The vector $t$ is used to generate the topology $\mathcal{T}$ with $\mathcal{T}_i = H(t_i)$, where:

$$H(t_i) = \begin{cases} 1 & \text{if } t_i > 0, \\ 0 & \text{if } t_i \leq 0. \end{cases}$$

Thus, if $t_i > 0$ then $\omega_i$ is active, otherwise (when $t_i \leq 0$), $\omega_i$ is pruned. We use a global penalty term $L_{-\infty}$ to push all $t_i$ values towards $-\infty$, which we discuss in detail in Section 3.2. Our goal is to find a topology $\mathcal{T}^*$ and set of weights $\omega^*$ such that the following loss is minimized:

$$\mathcal{J}(\omega, \mathcal{T}) = \mathcal{L}(\omega, \mathcal{T}) + L_{-\infty}(t).$$

## 3.2 Weight flux

We begin by introducing the notion of flux, evaluated on one topology $\mathcal{T}$, and develop its connection to weight importance. Since the optimal topology $\mathcal{T}^*$ is initially unknown, any metric measured on some topology $\mathcal{T}$ might not be relevant for $\mathcal{T}^*$. For this reason, we then extend flux to *aggregated flux*, a more informative evaluation based on a family of topologies $\mathcal{T}^{1 \to K}$.

We start by defining $\mathcal{G}_i(\omega, \mathcal{T})$, representing the direction in which $t_i$ needs to change to minimize the loss for topology $\mathcal{T}$ and weights $\omega$:

$$\mathcal{G}_i(\omega, \mathcal{T}) = -\frac{\partial \mathcal{L}(\omega, \mathcal{T})}{\partial t_i}, \forall t_i \in \mathbb{R}. \tag{1}$$

To allow computing (1) despite the non-differentiable step function $H(t_i)$, we employ a straight-through estimator for the gradient of $H$ with respect to $t_i$, which we denote by $\text{STE}_H$. Several choices for $\text{STE}_H$ will create the behavior we desire in $\mathcal{G}$ (e.g., $\text{STE}_H(t_i) = \sigma(t_i) \cdot (1 - \sigma(t_i))$, $\text{STE}_H(t_i) = 1 - \tanh^2(t_i)$), but none perform significantly better than the others in experiments. Therefore, for the sake of simplicity, we choose $\text{STE}_H(t_i) = 1$.

To fully understand the implications of $\mathcal{G}_i$ on updating $t_i$, we study the gradients composing it. We define $\theta_i = \omega_i \cdot H(t_i)$, and refer to $\theta_i$ as effective weight. By rewriting $\mathcal{G}_i$ we get:

$$\mathcal{G}_i(\omega, \mathcal{T}) = \underbrace{-\frac{\partial \mathcal{L}(\omega, \mathcal{T})}{\partial \theta_i}}_{=:\mathcal{A}_i} \cdot \frac{\partial \theta_i}{\partial t_i} = \mathcal{A}_i \cdot \omega_i \cdot \text{STE}_H(t_i) = \mathcal{A}_i \cdot \omega_i.$$

$\mathcal{A}_i$ represents the direction in which the effective weight $\theta_i$ should change to minimize the loss. If $\mathcal{A}_i$ has the same sign as the weight $\omega_i$, then $t_i$ will increase, reinforcing presence. Otherwise, if they have different signs, $t_i$ will decrease towards pruning. This behavior takes two meanings depending on whether $t_i \leq 0$ or $t_i > 0$, which we analyze below. For this purpose, we define $\mathcal{W}_i = -\frac{\partial \mathcal{L}}{\partial \omega_i}$, the direction in which $\omega_i$ should change to reduce the loss.

For $t_i > 0$, $\mathcal{W}_i = \mathcal{A}_i \cdot H(t_i) = \mathcal{A}_i$. Therefore, $\mathcal{G}_i(\omega, \mathcal{T})$ can be rewritten as $\mathcal{W}_i \cdot \omega_i$, meaning that $t_i$ increases when $\mathcal{W}_i$ and $\omega_i$ have the same sign and decreases otherwise. Note that $\mathcal{W}_i$ and $\omega_i$ having the same sign also means that $|\omega_i|$ increases, while opposite signs imply that $|\omega_i|$ decreases. Therefore, $t_i$ follows the direction of change in $|\omega_i|$.

To assess the importance of $\theta_i = \omega_i$, the method allows $t_i \leq 0$, causing $\theta_i = 0$, and checks whether as a result $\mathcal{A}_i$ points towards $\omega_i$, i.e. whether $\text{sign}(\mathcal{A}_i) = \text{sign}(\omega_i)$. If this is true, moving $\theta_i$ from $0$ towards $\omega_i$ would reduce the new loss (obtained after $\theta_i$ became 0) and consequently, $\mathcal{G}_i$ increases $t_i$ until regrowth, $\theta_i = \omega_i$. In this way, Hyperflux implements the key insight that *one never knows the value of something ($\theta_i$) until one loses it (sets it to 0)*. Otherwise, if $\text{sign}(\mathcal{A}_i) \neq \text{sign}(\omega_i)$, $t_i$ decreases, keeping the weight pruned, $\theta_i = 0$. All four combinations of signs are presented in Fig. 1. For this $t_i \leq 0$ setting, $\mathcal{G}_i$ takes the meaning of *flux*, and its relation to weight importance is further discussed in Appendix A.1.

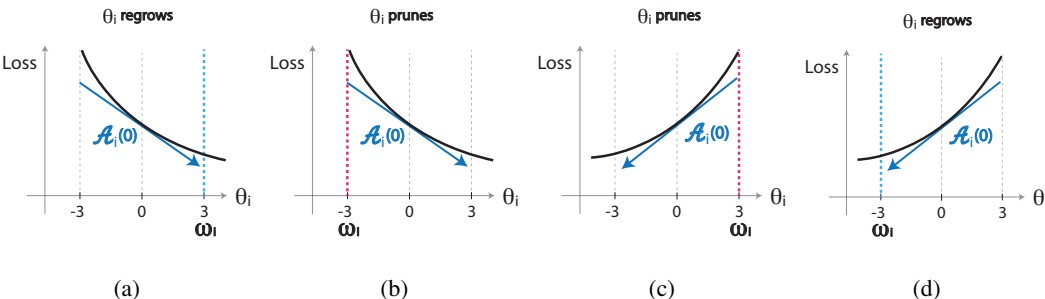

Figure 1: Scenarios for $\theta_i$ when $H(t_i) = 0$. If $\mathcal{A}_i$ points towards $\omega_i$ the flux $\mathcal{G}_i^-$ regrows the weight as in (a) and (d). Otherwise, it keeps the weight pruned as in (b) and (c). Numerical values are only illustrative.

Given the fact that $\mathcal{G}_i(\omega, \mathcal{T})$ takes two different meanings, we introduce two different notations:

$$\mathcal{G}_i(\omega, \mathcal{T}) =: \begin{cases} \mathcal{G}_i^-(\omega, \mathcal{T}), & t_i \le 0, \\ \mathcal{G}_i^+(\omega, \mathcal{T}), & t_i > 0. \end{cases} \tag{2}$$

$\mathcal{G}_i^-(\omega, \mathcal{T})$ refers to flux, whereas $\mathcal{G}_i^+(\omega, \mathcal{T})$ is the tendency of $|\omega_i|$.

Despite having flux as a metric of importance, we have not presented so far a criterion to prune the weights, that would lead us to uncover their flux. To drive $t$ values towards $-\infty$, we employ an "$L_{-\infty}$" loss called *pressure*, formulated as:

$$L_{-\infty}(t) = \frac{1}{d} \cdot \gamma \cdot \sum_{i=1}^{d} t_i, \tag{3}$$

where $\gamma$ is a scalar used to control sparsity and $d$ the total number of weights in the network. Any reference about an increase, decrease, value or scheduler of pressure will refer to $\gamma$. The pressure term yields a constant gradient $\frac{\gamma}{d}$ with respect to each $t_i$ parameter, independent of their current value.

We let $\mathcal{G}_i(\omega, \mathcal{T})$ and the gradient of $L_{-\infty}(t)$ interact during backpropagation without direct intervention. As a result, a family of topologies $\mathcal{T}^{1 \to K}$ *emerges implicitly* during training by concurrent pruning (determined by $L_{-\infty}$) and regrowth (determined by $\mathcal{G}_i^-$ increasing $t_i$). Furthermore, a $t_i \le 0$ may be increased for several iterations until it reaches $t_i > 0$, being evaluated at each iteration over a potentially different topology $\mathcal{T}^k \in \mathcal{T}^{1 \to K}$. This behavior is desirable, given that evaluating flux on a single topology provides a limited estimate of importance. To get a better picture of the underlying interactions, we begin by extending equation (2) to a family of topologies:

$$\mathcal{G}_i^{-/+}\left(\omega, \mathcal{T}^{1 \to K}\right) = \frac{1}{K} \sum_{k=1}^{K} \mathcal{G}_i^{-/+}\left(\omega, \mathcal{T}^k\right). \tag{4}$$

This leads to an *aggregated flux* $\mathcal{G}_i^-(\omega, \mathcal{T}^{1 \to K})$ and an average tendency of change in weight magnitude $\mathcal{G}_i^+(\omega, \mathcal{T}^{1 \to K})$ respectively. In Hyperflux, the updates over $H$ iterations write:

$$\sum_{h=1}^{H} \frac{\partial(\mathcal{L}(\mathcal{T}^h, \omega) + L_{-\infty}(t))}{\partial t_i} = \sum_{h=1}^{H} (\mathcal{G}_i(\omega, \mathcal{T}^h) + \frac{\gamma}{d}), \quad (5)$$

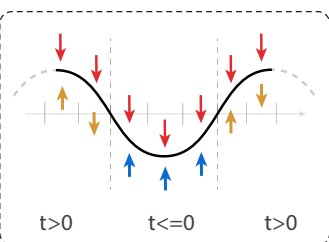

where $\mathcal{T}^h$ is the topology at iteration $h$. We examine the "life cycle" of a presence parameter $t_i$ over the $H$ training iterations. In figure 2 we show how the gradients of $t_i$, represented by arrows, interact. During these $H$ steps, $t_i$ alternates between active phases during which it follows tendency of $|\omega_i|$, and pruned phases during which flux accumulates. We refer to the transition from a pruned phase back to a present phase as *implicit regrowth*. To illustrate the interactions between flux and pressure in our method, consider a pruned phase beginning at iteration $P_s$ and ending at iteration $P_f$ ($1 < P_s < P_f \le H$). If $P_f$ marks the final step of that pruned phase, the total change in $t_i$ over $[P_s, P_f]$ is positive, which gives:

Figure 2: Depiction of gradients (as arrows) influencing $t_i$, red, yellow and blue denote pressure, $\mathcal{G}_i^+$ and respectively $\mathcal{G}_i^-$.

$$\sum_{h=P_s}^{P_f} (\mathcal{G}_i(\omega, \mathcal{T}^h) - \frac{\gamma}{d}) > 0 \iff (P_f - P_s) \cdot \left[\mathcal{G}_i^-\left(\omega, \mathcal{T}^{P_s \to P_f}\right) - \frac{\gamma}{d}\right] > 0. \tag{6}$$

Thus, a weight will be regrown if the aggregated flux is greater than the pressure. Conversely, after an active interval, the weight becomes pruned i.e. $\left[\mathcal{G}_i^+\left(\omega, \mathcal{T}^{P_s \to P_f}\right) - \frac{\gamma}{d}\right] < 0$. This mechanism influences all weights: pressure pushes them toward pruning, but they regrow whenever the aggregated flux exceeds that pressure. Consequently, since our method relies on weights that already encode meaningful information, we begin pruning by initializing the network with *pretrained weights*.

### 3.3 Pressure & Flux Properties

Following from the theoretical insights about flux and pressure described so far, we postulate a series of properties that naturally emerge from these concepts. We experimentally validate each one of the properties, confirming our expectations, and laying the foundation for our $\gamma$ scheduler.

**Property 1: Sparsity Convergence for a Fixed** $\gamma$. As sparsity increases and the number of weights decreases, fewer weights are used to represent the same information contained within the dataset, so the overall importance and flux of the remaining weights should be larger. Once the flux of the remaining weights surpasses the pressure, sparsity should converge. Therefore, we ask the following question: *Given a fixed $\gamma$, will the network converge to a final sparsity $\mathcal{S}$?* In Figure 3a, we test this by running LeNet-300 on MNIST and ResNet-50 on Cifar-10. We allow each network to train for 300 to 1000 epochs with a constant pressure $\gamma$ and observe the results.

We test two different optimizers for $t$ values, SGD and Adam, while for weights we use the same Adam optimizer everywhere (more on training setup and its notation in Appendix F). Our findings suggest that there is no one curve that fits the decrease in parameters for both optimizers, but $\mathcal{S}$ is the same regardless of the optimizer used. An important observation is that $\mathcal{S}$ is influenced by the weights learning rate $\eta_\omega$. If $\eta_\omega$ is high, convergence happens in a larger number of epochs (1000 in our experiments), at a higher sparsity. If $\eta_\omega$ is low, convergence happens sooner (300 epochs), at a lower sparsity. One way to ensure smooth convergence is to decrease $\eta_\omega$ during training. Otherwise, the network tends to converge more slowly, as seen in the green curve experiment. Further ablation studies are found in Appendix B.

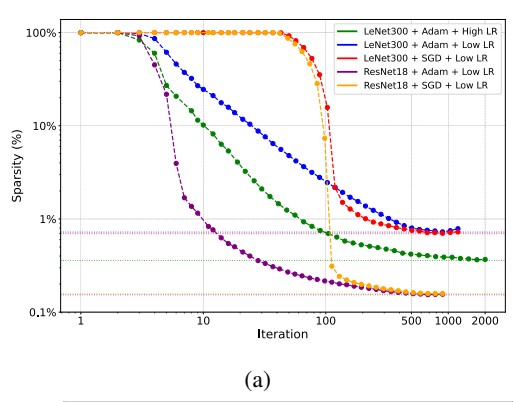

(a)

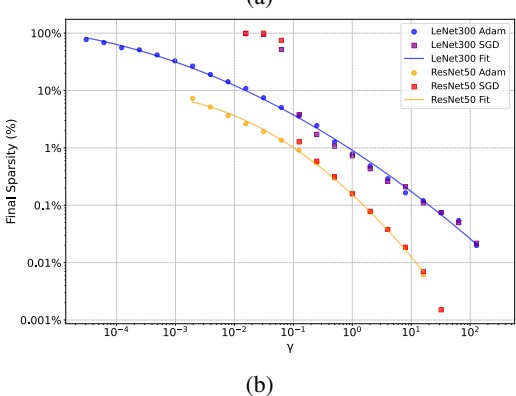

(b)

Figure 3: Convergence for fixed $\gamma = 2$ is showcased in (a), while in (b) we present the relationship between $\gamma$ and final sparsity.

**Property 2: Relationship Between $\gamma$ and Final Sparsity**. Assuming as illustrated above that all networks have a sparsity they converge to for a fixed $\gamma$, we ask: *Can we find the relationship between $\gamma$ and $\mathcal{S}$?* We modify the previous experiment to run the networks for 300 epochs with the same training setup for several values of $\gamma$. Our empirical results suggest a generalized scaling law:

$$\ln(s) = \ln(c) - \alpha_0 \ln(\gamma) - \alpha_1 \left(\ln(\gamma)\right)^2 \quad (7)$$

where constants, c, $\alpha_0, \alpha_1$ depend on dataset, network architecture and training setup. Figure 3b showcases different convergence points for different optimizers and $\gamma$ values. The curves bend more sharply toward the end as the network loses accuracy (and feature representations), yielding a lower convergence point, until the network collapses, pruning all weights. We call this property by the name *Neural Pruning Law*.

**Property 3: Important weights developing large flux** is probably the most important idea in Hyperflux. Therefore we ask: *How large is the flux of critical weights compared to other weights?* To obtain a set of critical weights, we create a bottleneck in a LeNet-300 network by pruning only the last weight matrix, until an identity remains between the hidden layer and each unit of the output layer (in our case 10 weights). We measure their flux by reporting the largest pressure that still does not prune the weights, since we know that weights carrying greater flux demand higher pressure to prune, see equation (6).

As a baseline of comparison, we use the original network without the bottleneck, and report several $\gamma$ and the corresponding $\mathcal{S}$ they produce. The

Table 1: Pressure needed to prune the bottleneck

| Sparsity (%) | Pressure |
|---|---|
| 99.05 | $\gamma_B$ |
| 99.75 | $\gamma_B \cdot 2^3$ |
| 99.95 | $\gamma_B \cdot 2^7$ |
| Bottleneck | $\gamma_B \cdot 2^{13}$ |

results are reported in Table 1. The bottleneck weights have $2^{13}$ more flux than the weights of a 99% sparsified network.

## 3.4 Pressure Scheduler & Stabilization Stage

Our findings from Section 3.3 suggest that a $\gamma$ exists for any desired $\mathcal{S}$. However, in practical applications, $\gamma$ is not known at the start and tuning it would require hyperparameter search. Instead, we propose a dynamic scheduler that adjusts $\gamma$ after each epoch automatically, driving the network towards a desired sparsity. Furthermore, to ensure convergence of weights after pruning, we introduce a stabilization stage at the end.

**Pressure Scheduler:** The goal of our scheduler is to adjust $\gamma$ such that the network converges to $\mathcal{S}$ with minimal accuracy decrease. We denote by $\gamma_e$ and $s_e$, the pressure and network sparsity at epoch $e$. Because the frequency of updates is constant, occurring after each epoch, any non-linear change in pressure required to affect sparsity (see Eq. 7) must arise from the update rule. To get this nonlinearity we set $\gamma_e = (p_e)^\alpha$, with $p_e$, a scalar base, updated according to Algorithm 1. Inertia terms $p_+$ and $p_-$ account for suboptimal $\alpha$ or $u$. Apart from non-linear updates, our scheduler requires a binary pressure policy $\Pi$ to determine when those updates are applied such that $\mathcal{S}$ is reached. We explore two choices for $\Pi$. In the first, $s_e$ follows a user-defined curve $f(e)$, trading precise convergence to $\mathcal{S}$ for trajectory control. In this case $\Pi(E_p, \mathcal{S}, s_e, e)$ is true if and only if $s_e < f(e)$.

---

**Algorithm 1** Pressure scheduler - SCHED($s_e, e$)

1: **Input:** Current sparsity $s_e$ and epoch $e$
2: **Requires:** Pruning epochs $E_p$, desired final sparsity $\mathcal{S}$, pressure policy $\Pi$, step $u$, exponent $\alpha$.
3: **Internals:** Positive and negative inertia $p_+$, $p_-$, base scalar $p_e$ for epoch 1, $p_1$ (all initialized to 0).
  $\triangleright$ *Runs after each epoch*
4: **if** $\Pi(E_p, \mathcal{S}, s_e, e)$ **then**
5:   $p_e \leftarrow p_{e-1} + u + p_+$
6:   $p_+ \leftarrow p_+ + \frac{u}{4}$
7:   $p_- \leftarrow 0$
8: **else**
9:   $p_e \leftarrow p_{e-1} - u - p_-$
10:   $p_- \leftarrow p_- + \frac{u}{4}$
11:   $p_+ \leftarrow 0$
12: **end if**
13: **Return:** pressure $\gamma_e = (p_e)^\alpha$

---

In the second policy, $s_e$ stays between a dynamic upper bound and the target $\mathcal{S}$, achieving precise convergence to $\mathcal{S}$ at the cost of poorer trajectory control. Both policies are discussed in Appendix E.

**Stabilization Stage:** One side effect of Hyperflux is the noise created by pruning and reactivation of weights, which while helpful for pruning, is harmful for convergence. For this reason, to allow the weights and network topology to converge, we introduce a stabilization stage. Specifically, we set the pressure to zero to encourage regrowth while simultaneously decaying the learning rate $\eta_t$ to prevent excessive reactivation.

---

**Algorithm 2** Hyperflux Pruning Algorithm

1: **Input:** Pretrained weights $\omega^{\text{init}}$, pruning epochs $E_p$, stabilization epochs $E_s$ (leading to total epochs $E_t = E_p + E_s$), pressure scheduler SCHED($s_e, e$).
2: **Output:** Weights $\omega^*$, final topology $\mathcal{T}^*$.
3: **Initialize:**
4:   Weights $\omega \leftarrow \omega^{\text{init}}$.
5:   Presence parameters $t_i \leftarrow$ positive values, $\forall i \in \{1, 2, ..., d\}$.
6:   Topology $\mathcal{T}_i \leftarrow 1, \forall i \in \{1, 2, ..., d\}$.
7: **for** epoch $e = 1$ to $E_t$ **do**
8:   Calculate total loss $\mathcal{J}(\omega, \mathcal{T}) = \mathcal{L}(\omega, \mathcal{T}) + L_{-\infty}(t)$.
9:   $\omega \leftarrow \omega - \eta_\omega \nabla_\omega \mathcal{L}$.
10:   $t \leftarrow t - \eta_t \nabla_t \mathcal{J}$.
11:   **if** $e \leq E_p$ **then**
12:     $\gamma \leftarrow$ SCHED(current sparsity $s_e, e$)
13:   **else**
14:     $\eta_t \leftarrow 0.9 \cdot \eta_t$
15:     $\gamma \leftarrow 0$
16:   **end if**
17: **end for**

---

## 4 Performance Comparison

To validate *Hyperflux*, we conduct comprehensive pruning experiments on a diverse set of architectures and datasets: ResNet-50 and VGG-19 on CIFAR-10/100, and ResNet-50 on ImageNet-1K. We pit Hyperflux against state-of-the-art pruning approaches such as GraNet [30], GMP [59], Spartan [44], and AC/DC [36]. To ensure a fair comparison, we run ourselves all other methods, initializing them with the pretrained weights used in Hyperflux, while maintaining the same training budget and augmentations. We test several training setups for each method and report the best results, to ensure no unfair degradation occurs due to suboptimal hyperparameters.

Additionally, to better position Hyperflux within the broader literature, we choose to include one-shot methods [24, 49, 45] commonly used as benchmarks in other works, even though our post-training setup is not applicable to them. These benchmarks will be marked with $*$.

None of our comparison methods incorporate learnable masks as Hyperflux does. Although we identified some mask-based methods [40, 32, 57], their differences in benchmarks, methodology or missing code prevent a direct comparison to our work. Each configuration but ResNet-50 on ImageNet is run three times and we report the results as mean $\pm$ standard deviation, all experiments are run on three NVIDIA GeForce RTX 4090 GPUs. Full details on training recipe are in Appendix F.

### 4.1 CIFAR-10 / 100

We evaluate the performance of *Hyperflux* on CIFAR-10 and CIFAR-100 using ResNet-50 and VGG-19 architectures. Results are presented in Table 2. On CIFAR-10, *Hyperflux* outperforms the baseline at 90%, 95%, and 98% sparsity for both VGG-19 and ResNet-50, with accuracy gains under 1% over the next best. Specifically, for VGG-19, it beats GraNet by $0.18\%$ and GMP by $0.23\%$ at 90% sparsity (rising to $1.61\%$ over GMP at 98%), while on ResNet-50 it maintains a $0.7\%$ lead over GraNet across all levels. We also analyze ResNet-50's layer-wise sparsity at extreme rates (99.74%, 99.01%, 98.13%) and illustrate weight distribution changes in Appendix C.1.

On CIFAR-100, *Hyperflux* leads in 4 of 6 benchmarks, being behind GraNet by only $0.1\%$ and $0.3\%$ in the other two. Notably, GraNet gains nearly $2\%$ on ResNet-50 when initialized with our pretrained weights. Conversely, RigL gains $1.5\%$ points of accuracy on ResNet-50 for CIFAR-100, yet experiences drops of up to $0.3\%$ on ResNet-50 for CIFAR-10. On the remaining two benchmarks, its gains are only moderate. At 90% and 95% sparsity, *Hyperflux* outperforms all methods, including GraNet, by $0.5\%$. Furthermore, GMP finds itself at a difference of $0.2\%$ at 98% sparsity on VGG-19, increasing to $1.2\%$ points of accuracy at 90% sparsity, while RigL is behind by $2.9\%$ at 98% and $1.3\%$ at 90% sparsity.

Table 2: Comparison on CIFAR-10 and CIFAR-100 datasets at different pruning ratios (90.0%, 95.0%, 98.0%). Bold values represent the best performance for each setting.

| Dataset | CIFAR-10 | | | CIFAR-100 | | |
|---|---|---|---|---|---|---|
| Pruning ratio | 90.0% | 95.0% | 98.0% | 90.0% | 95.0% | 98.0% |
| **VGG-19** (Dense) | | $93.85 \pm 0.06$ | | | $73.44 \pm 0.09$ | |
| SNIP* | 93.63 | 93.43 | 92.05 | 72.84 | 71.83 | 58.46 |
| GraSP* | 93.30 | 93.04 | 92.19 | 71.95 | 71.23 | 68.90 |
| Synflow* | 93.35 | 93.45 | 92.24 | 71.77 | 71.72 | 70.94 |
| GMP | $93.82 \pm 0.15$ | $93.84 \pm 0.14$ | $92.34 \pm 0.13$ | $73.57 \pm 0.20$ | $73.39 \pm 0.11$ | $72.78 \pm 0.07$ |
| RigL | $93.60 \pm 0.15$ | $93.17 \pm 0.09$ | $92.39 \pm 0.04$ | $73.03 \pm 0.14$ | $72.68 \pm 0.22$ | $70.02 \pm 0.7$ |
| GraNet ($s_i = 0$) | $93.87 \pm 0.05$ | $93.84 \pm 0.16$ | $93.87 \pm 0.11$ | $74.08 \pm 0.10$ | $73.86 \pm 0.04$ | $\textbf{73.00} \pm \textbf{0.18}$ |
| Hyperflux (ours) | $\textbf{94.05} \pm \textbf{0.17}$ | $\textbf{94.15} \pm \textbf{0.14}$ | $\textbf{93.95} \pm \textbf{0.18}$ | $\textbf{74.37} \pm \textbf{0.21}$ | $\textbf{74.18} \pm \textbf{0.15}$ | $72.9 \pm 0.05$ |
| **ResNet-50** (Dense) | | $94.72 \pm 0.05$ | | | $78.32 \pm 0.08$ | |
| SNIP* | 92.65 | 90.86 | 87.21 | 73.14 | 69.25 | 58.43 |
| GraSP* | 92.47 | 91.32 | 88.77 | 73.28 | 70.29 | 62.12 |
| Synflow* | 93.35 | 93.45 | 92.24 | 71.77 | 71.72 | 70.94 |
| RigL | $94.02 \pm 0.33$ | $93.76 \pm 0.23$ | $92.93 \pm 0.1$ | $78.04 \pm 0.19$ | $77.39 \pm 0.21$ | $75.61 \pm 0.11$ |
| GMP | $94.81 \pm 0.05$ | $94.89 \pm 0.1$ | $94.52 \pm 0.12$ | $78.39 \pm 0.18$ | $78.38 \pm 0.43$ | $77.16 \pm 0.25$ |
| GraNet ($s_i = 0$) | $94.69 \pm 0.08$ | $94.44 \pm 0.01$ | $94.34 \pm 0.17$ | $79.09 \pm 0.23$ | $78.71 \pm 0.16$ | $\textbf{78.01} \pm \textbf{0.20}$ |
| Hyperflux (ours) | $\textbf{95.41} \pm \textbf{0.12}$ | $\textbf{95.15} \pm \textbf{0.11}$ | $\textbf{95.26} \pm \textbf{0.13}$ | $\textbf{79.58} \pm \textbf{0.18}$ | $\textbf{79.23} \pm \textbf{0.16}$ | $77.7 \pm 0.08$ |

## 4.2 ImageNet-2012

To test *Hyperflux* at scale, we pruned ResNet-50 on ImageNet-2012. Table 3 shows that, even at extreme sparsity, *Hyperflux* performs competitively against state-of-the-art. Interestingly, our loading of pretrained weights increased the accuracy of all methods, with the exception of Spartan, which lost almost 1.5% accuracy compared to its reported results.

At 96.42% sparsity, *Hyperflux* reaches 72.21% accuracy, surpassing GMP, GraNet and Spartan, while performing competitively against AC/DC, at a difference of 0.3%. This hierarchy is maintained for both 90% and 95% sparsity, with the gap between Hyperflux and AC/DC remaining below 0.6 points in accuracy. We conducted an analysis on the weight histograms of ResNet-50 on ImageNet to study the difference in weight distribution and observed that Hyperflux pruned aggressively the convolutional layers, details in Appendix C.1

The computational cost is only assessed on ImageNet-1k as it is the most intensive benchmark. Pruning cuts FLOPs to 0.15× inference/0.60× training at 90% sparsity, and 0.08×/0.52× at 95% sparsity. Despite incurring larger costs for training than other methods, Hyperflux is able to produce sparse networks whose inference cost is lower. This is caused by the per-layer sparsity distribution generated by our method, which prunes more the layers contributing most to the computational cost. For the baselines, we report the computational costs when they are available in their respective papers, and fill with $-$ when they are not. More details on computational cost are given in Appendix D.

Table 3: ResNet-50 top-1 accuracy, parameter count, sparsity, and compute cost on ImageNet-2012. We denote by $s$ the sparsity, and by $F_{\text{train}}$ and $F_{\text{test}}$ the compute cost (FLOPs) required for training and testing, respectively.

| Method | Top-1(%) | Params | s(%) | $F_{\text{test}}$ | $F_{\text{train}}$ |
|---|---|---|---|---|---|
| ResNet-50 | 77.01 | 25.6M | 0.00 | 1.00× | 1.00× |
| GMP | 74.29 | 2.56M | 90.00 | 0.10× | 0.51× |
| GraNet | 74.68 | 2.56M | 90.00 | 0.16× | 0.23× |
| Spartan | 75.12 | 2.56M | 90.00 | 0.14× | - |
| AC/DC | 75.83 | 2.56M | 90.00 | 0.18× | 0.58× |
| **Hyperflux** | 75.28 | 2.54M | 90.11 | 0.15× | 0.60× |
| GMP | 70.95 | 1.28M | 95.00 | 0.05× | - |
| GraNet | 72.83 | 1.28M | 95.00 | 0.12× | 0.17× |
| Spartan | 72.92 | 1.28M | 95.00 | 0.08× | - |
| AC/DC | 74.03 | 1.28M | 95.00 | 0.11× | 0.53× |
| **Hyperflux** | 73.30 | 1.28M | 95.00 | 0.08× | 0.52× |
| GMP | 70.62 | 0.90M | 96.50 | - | - |
| GraNet | 71.06 | 0.90M | 96.50 | 0.09× | 0.15× |
| Spartan | 71.13 | 0.90M | 96.50 | - | - |
| AC/DC | 72.50 | 0.90M | 96.50 | - | - |
| **Hyperflux** | 72.21 | 0.92M | 96.42 | 0.06× | 0.49× |

## 5 Conclusions, Limitations and Future Work

We introduced Hyperflux, a conceptually grounded $L_0$ method in which we construct the notions of flux and pressure and study their relationship with weight importance. Furthermore, we postulate and validate several properties of Hyperflux that enhance its explainability. Finally, our experiments show strong performance compared to existing state-of-the-art methods.

Despite its advantages, Hyperflux has several areas which could be improved. Our method incurs at least 33% of the dense network's computational cost (see Appendix D) and demands additional hyperparameters (e.g. scheduler policy and step, $\eta_t$) which work well at the same values across the vision tasks we tested on, but may require adjustment on other tasks. To address some of these issues, we can treat the network sparsity as the output of a dynamical control problem and the pressure as its input, so as to tightly control the transient and steady-state sparsity $\mathcal{S}$ despite differences in the tasks. Additionally, we are interested in checking whether the empirical Neural pruning law we found generalizes to other deep learning tasks.

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

## A  Analysis

### A.1  Why Important Weights Generate Stronger Flux

To study the flux of important weights, let us focus on a specific weight $\omega_i$ in the regime $t_i \leq 0$, and thus $\theta_i = 0$. For analytical purposes, we define the loss in terms of the effective weights $\theta_i = \omega_i \cdot H(t_i)$ as $\mathcal{L}(\theta)$, where $\mathcal{L}(\theta|\theta_i = 0)$ is the loss when $t_i \leq 0$ and $\mathcal{L}(\theta|\theta_i = \omega_i)$ is the loss when $t_i > 0$. We perform a Taylor expansion of $\mathcal{L}(\theta)$ around $\theta_i = 0$. By perturbing $\theta_i$ by $\omega_i$ (i.e. by approximating the effect of regrowing the weight), we observe that the first-order term in the expansion is the flux of $\omega_i$. Formally:

$$\mathcal{L}(\theta|\theta_i = \omega_i) = \mathcal{L}(\theta|\theta_i = 0) + \omega_i \frac{\partial \mathcal{L}(\theta|\theta_i = 0)}{\partial \theta_i} + \frac{1}{2}\omega_i^2 \frac{\partial^2 \mathcal{L}(\theta|\theta_i = 0)}{\partial \theta_i^2} + O(\omega_i^3). \qquad (8)$$

Recalling the formula for flux, $\mathcal{G}_i^-(\omega, \mathcal{T})$, and neglecting the second and higher-order terms:

$$\mathcal{L}(\theta|\theta_i = 0) - \mathcal{L}(\theta|\theta_i = \omega_i) \approx -\omega_i \frac{\partial \mathcal{L}(\theta|\theta_i = 0)}{\partial \theta_i} = \mathcal{G}_i^-(\omega, \mathcal{T}).$$

Thus we obtain a direct relationship between flux and weight importance: the flux approximates the change in the loss that could be incurred when the weight is regrown. However, this relationship holds only up to neglected higher-order terms, so it should be viewed as a useful approximation rather than an exact law.

### A.2  Flux Connection To The Hessian

To relate flux to other importance metrics, specifically the Hessian, we consider the Taylor approximation from (8) and write:

$$\mathcal{L}(\theta|\theta_i = 0) - \mathcal{L}(\theta|\theta_i = \omega_i) = -\left( \omega_i \frac{\partial \mathcal{L}(\theta|\theta_i = 0)}{\partial \theta_i} + \frac{1}{2}\omega_i^2 \frac{\partial^2 \mathcal{L}(\theta|\theta_i = 0)}{\partial \theta_i^2} \right) - O(\omega_i^3).$$

Given that the flux $\mathcal{G}_i^-(\omega, \mathcal{T}) = -\omega_i \frac{\partial \mathcal{L}(\theta|\theta_i=0)}{\partial \theta_i}$ and neglecting terms of order $O(\omega_i^3)$ and higher, we obtain:

$$\mathcal{L}(\theta|\theta_i = 0) - \mathcal{L}(\theta|\theta_i = \omega_i) \approx \mathcal{G}_i^-(\omega, \mathcal{T}) - \frac{1}{2}\omega_i^2 \underbrace{\frac{\partial^2 \mathcal{L}(\theta|\theta_i = 0)}{\partial \theta_i^2}}_{H_{ii}^\theta}$$

The second term, $-\frac{1}{2}\omega_i^2 H_{ii}^\theta$, contains the diagonal element, $H_{ii}^\theta$, of the Hessian matrix of the loss function with respect to $\theta_i$. This shows that our flux metric captures the linear component of the loss change, while the second term captures the quadratic component, which is generally associated with Hessian-based pruning methods like Optimal Brain Damage [22]. In Optimal Brain Damage, a weight's saliency is estimated by $\frac{1}{2}H_{ii}\omega_i^2$, typically under the assumption that the network is at a minimum where first-order gradients are zero. On the other hand, Hyperflux prunes the weights, therefore recovering the first linear component of the Taylor expansion which becomes 0 when weights converge.

## B  Ablation Studies

### B.1  Factors influencing flux $\mathcal{G}_i^-(\omega, \mathcal{T})$

We begin by analyzing how the flux value $\mathcal{G}_i^-(\omega, \mathcal{T})$ is influenced by factors other than $\eta_t$, the learning rate on presence parameters. Our findings from Section 3.3, suggest that weight learning affects the behavior of flux, by changing the final convergence point a network will reach for the same constant pressure $\gamma$. We study this effect in the case of LeNet-300. We run the network for 1000 epochs for three different learning rates of $0.005, 0.0005$ and $0.00005$, with no schedulers used and the same constant $\gamma$. Our findings are summarized in Figure 4, which shows that increasing $\eta_\omega$, the weights learning rate, leads to smaller fluxes and convergence at higher sparsities.

Given the impact of $\eta_\omega$ on network convergence, we study the influence of high and low learning rates on our pruning and regrowth phases. In our experiments, we study three setups on ResNet-50 with CIFAR-10. In the first two experiments, we study how constant learning rates across the entire pruning and regrowth process affect sparsity and regrowth. We choose a high learning rate of 0.01 and a low learning rate of 0.0001. For our third experiment, we start with the high learning rate which is then decayed using cosine annealing to a low learning rate until the end of regrowth. For all three studies we let our scheduler guide the network towards the same sparsity rate of 1%. However, we observe significant differences in the regrowth stage. For the first experiment, regrowth

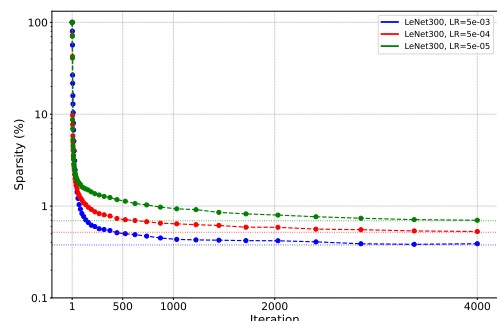

Figure 4: MNIST convergence for constant $\theta = 1$ for different learning rates

does not occur at all, with more weights being pruned even after the pressure is set to 0, while for the low learning rate, the performance initially degrades, but is followed by a substantial regrowth stage where the number of remaining parameters increases by 60%. For the third experiment performance does not degrade as much as for the low learning rate and the regrowth is done in a more controlled way, experiencing an increase in remaining parameters of 35%. The results are illustrated in Figure 5.

Lastly, we study how weight flux is affected by weight decay. Being directly applied on the weights, weight decay acts on both pruned and present weights. If a weight has been pruned in the first epochs on the training, weight decay will keep making it smaller and smaller, in this way diminishing its flux. We run similar experiments to the ones before, with a learning rate of 0.01, decayed during training to 0.0001, both with and without the standard weight decay. As expected, we observe in Figure 7a that regrowth without weight decay is more ample. We run this experiment five times, and note that each time the pattern illustrated in the figure remains consistent.

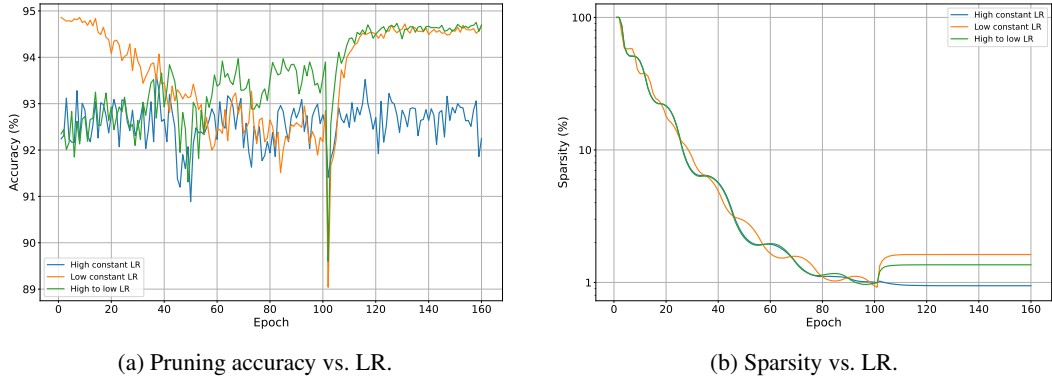

(a) Pruning accuracy vs. LR.

(b) Sparsity vs. LR.

Figure 5: The impact of the weights' learning rate on pruning accuracy (left) and achieved sparsity (right).

## B.2 Weights $\eta_\omega$ and pruning

Given the large impact $\eta_\omega$ has on flux, we explore its implications for producing an optimal pruning setup for Hyperflux. We run three experimental setups on ResNet-50 CIFAR-10 similar to the ones before. For each one of them, we select a starting learning rate, which is then decayed during training to 0.0001 to ensure convergence. For this setup, we run experiments using $\eta_\omega = 0.1, 0.01, 0.0001$. We analyze the results from the perspective of accuracy after pruning, noise, regrowth, and final accuracy. We find that the third setup is the most effective for Hyperflux.

We observe that each of the four studied aspects has a relationship with the learning rate. The noise is increased as initial learning rate increases, accuracy at the end of pruning is decreased the most for low learning rates and the highest for large learning rates. We obtain the highest final accuracy

for higher learning rates and the regrowth phase is diminished the higher the learning rate. These relationships hold and can be easily seen in Figure 7.

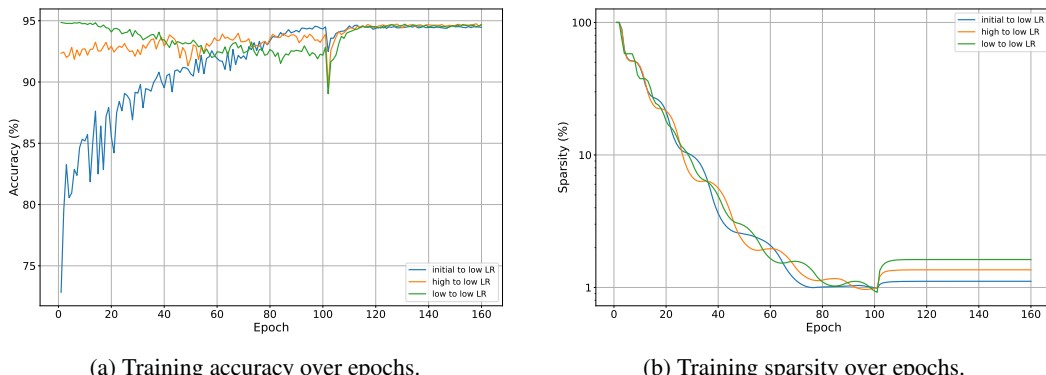

(a) Training accuracy over epochs.

(b) Training sparsity over epochs.

Figure 6: Training evolution for different learning rate configurations.

### B.3 $\eta_t$ values and regrowth

We analyze regrowth behavior for several values of $\eta_t$. At regrowth stage, we scale $\eta_t$ with $5, 10, 20, 30$ for VGG-19 on CIFAR-100 to observe the behavior of regrowth stage. Our findings are summarized in Figure 7b. As $\eta_t$ increases so does the number of regrown weights. However, we note that after a point, generally about an increase of $50\%$ in remaining parameters, the effects of regrowth start to be diminished and starts introducing noise in the performance, while also regrowing more weights.

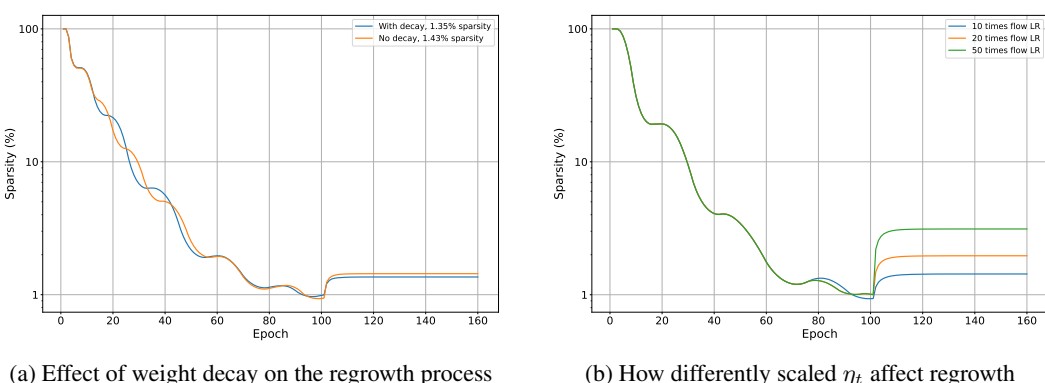

(a) Effect of weight decay on the regrowth process

(b) How differently scaled $\eta_t$ affect regrowth

Figure 7: Factors affecting regrowth.

## C  Extended experiments

### C.1  Layerwise sparsity levels & Weight Histograms

In this section, we examine the layer-wise sparsity observed for ResNet-50 on CIFAR-10 across the following pruning rates: 99.74%, 99.01%, and 98.13%. As illustrated in Figure 8, the overall sparsity hierarchy is maintained, displaying a decreasing trend in sparsity from the initial layers down to the final layer, where this pattern is interrupted. We hypothesize that earlier layers retain more weights due to their critical role in feature extraction, while deeper layers can sustain higher levels of pruning without significantly impacting overall performance. Notably, the penultimate layer experiences the highest degree of pruning, which means that it contains higher redundancy or less critical weights for performance. Furthermore, by analyzing the weight histograms for ResNet-50 with sparsity levels of 99.01% and 99.74% in Figure 11, we observe the influence of sparsity on the weight distributions.

High sparsity levels significantly alter weight distributions, demonstrating that extreme pruning not only reduces the number of active weights but also changes the underlying weight dynamics within the network.

The histograms in Figure 12 illustrate the differences in weight distributions between the pruning and regrowth stages on ImageNet with ResNet-50 at approximately 4.23% remaining weights. In the pruning stage, weights are more evenly distributed across the range of $[-0.4, 0.4]$, with a noticeable dip near zero, reflecting the removal of low-magnitude weights. In contrast, during regrowth stage the weight distribution shifts significantly, showing a sharp clustering of weights around zero, indicating the reactivation of low-magnitude weights during this process. This change in distribution correlates with a notable performance gap: the regrowth stage achieves 72.4% accuracy, while the pruning stage reaches only 66.13%, we consider the cause of this to be the fact that during the pruning process the small magnitude weights are pruned and during the regrowth phase we recover from these weights the ones that improve performance the most.

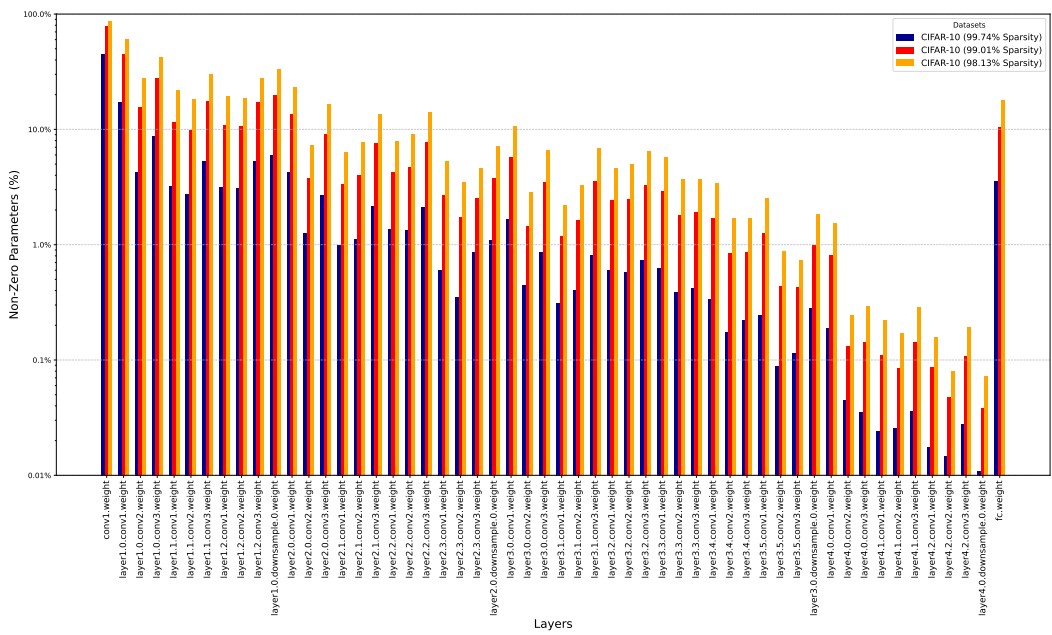

Figure 8: Per-layer sparsity for ResNet-50 CIFAR-10. We present 3 levels of sparsity: $99.74\%, 99.01\%$ and $98.13\%$.

## C.2 Implicit regrowth

Implicit regrowth serves as the main source of noise in our network, promoting diverse topologies throughout the training process. In Figure 9, we identify patterns in flip frequency, such as the lower number of flips at the start of training. This behavior is anticipated, as pruning a critical weight early on allows its features to be more readily absorbed by other weights. Around iteration 14, we notice a plateau followed by a brief decline in weight flips, which we attribute to the network stabilizing during this phase.

As training progresses and the number of parameters declines, the per-weight flip frequency continues to increase, while the overall flip frequency remains relatively steady, resulting in a continue increase of the per-weight flip frequency. The regrowth phase is marked by a sharp decrease in the total number of flips as the network stabilizes and the learning rate of flux parameters diminishes toward zero. This pattern is visible between iterations 70 and 130, alongside a gradual increase in the number of parameters.

In Figure 9 we can observe the behavior of *flux* in relation to the gradients of $t$ values. Note that negative values of the gradients translate into positive updates for $t$ values and vice-versa.Two specific type of weights emerge, the first type can be seen in the top-left and bottom-right diagrams in Figure 10, where the gradient $\mathcal{G}_i^+(\omega, \mathcal{T})$ does not oppose significant pressure for $t_i > 0$. This

leads to the weight being pruned multiple times, which coincides, with large negative values in the gradient, which push $t_i$ back over 0. The second type of weights, as common as the first one, does not get pruned at all. In this case, $\mathcal{G}_i^+(\omega, \mathcal{T})$ averaged over several iterations, attempts to increase the magnitude of the weight, therefore increasing $t_i$ at the same time, which leads to the weight not being pruned at all. We can see that in this case the overall magnitude of the gradients is below $-1.5$, which in our experiment was enough to resist pressure.

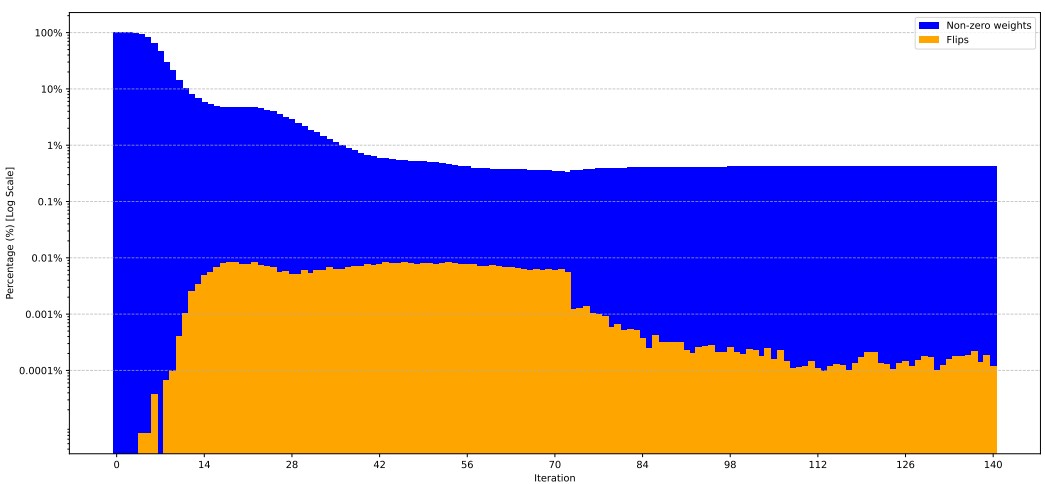

Figure 9: Frequency of Flips: The blue histogram represents the percentage of remaining parameters on a logarithmic scale, while the orange histogram illustrates the ratio of parameter flips per iteration relative to the total number of network parameters, also on a logarithmic scale. In our figure, one iteration is equivalent to the aggregation of 100 actual training iterations. We aggregate iterations to present the flip data in a more manageable way.

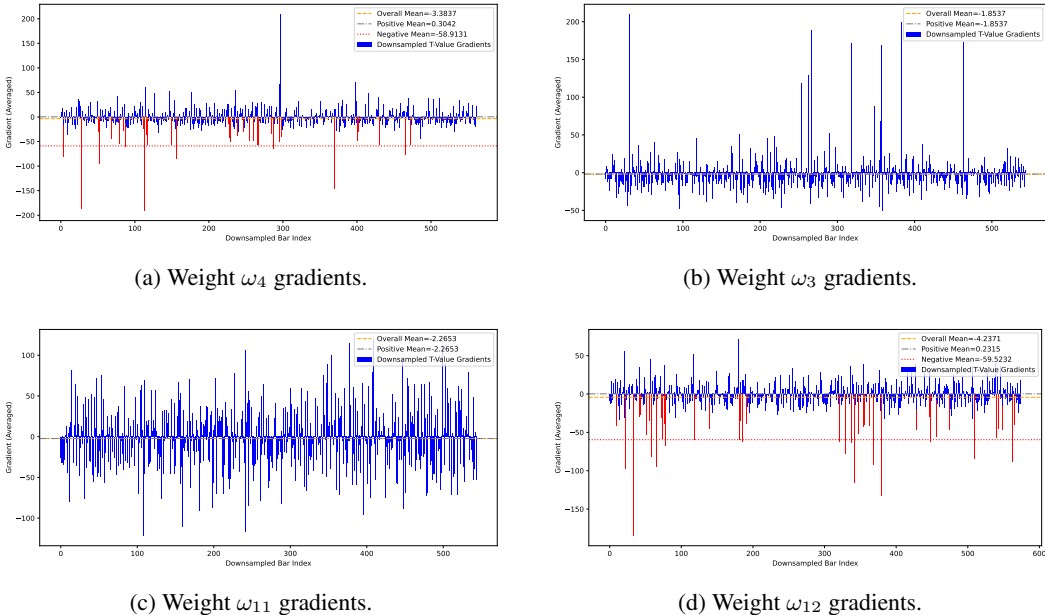

(a) Weight $\omega_4$ gradients.

(b) Weight $\omega_3$ gradients.

(c) Weight $\omega_{11}$ gradients.

(d) Weight $\omega_{12}$ gradients.

Figure 10: Gradient values over time for four remaining weights in the pruned network. Blue bars show gradients when $t_i > 0$, red when $t_i \leq 0$. Notice the high-magnitude red gradients (flux $\mathcal{G}_i^-(\omega, \mathcal{T})$) versus the typically smaller positive gradients (momentum $\mathcal{G}_i^+(\omega, \mathcal{T})$).

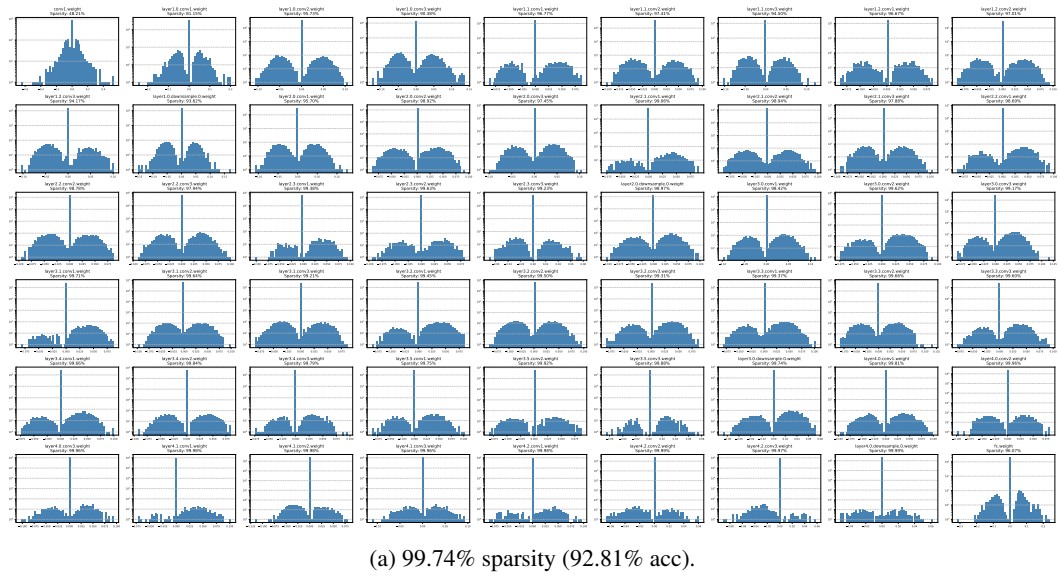

(a) 99.74% sparsity (92.81% acc).

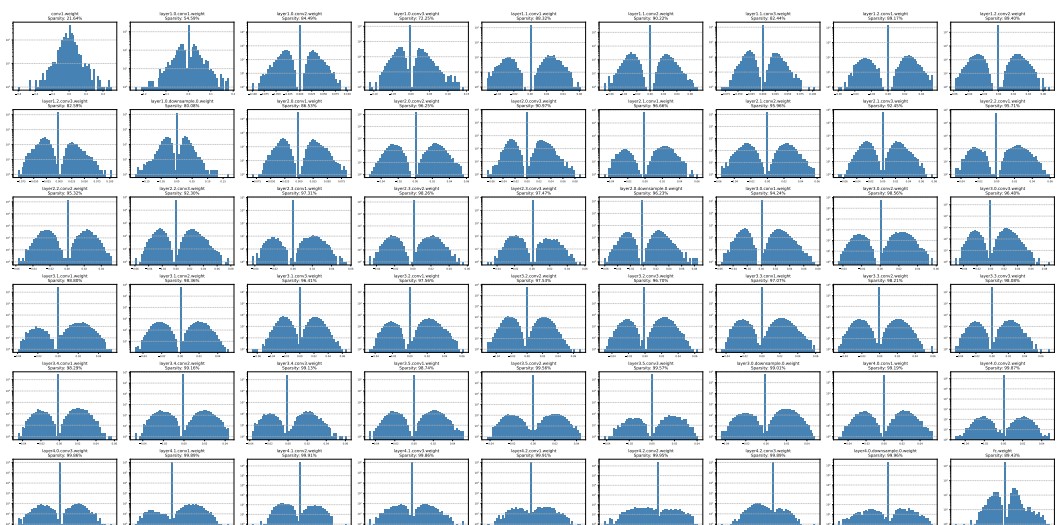

(b) 99.1% sparsity (94.44% acc).

Figure 11: Weight-value histograms of ResNet-50 on CIFAR-10 at two different sparsity levels. Note how the weight distribution reshapes as sparsity increases.

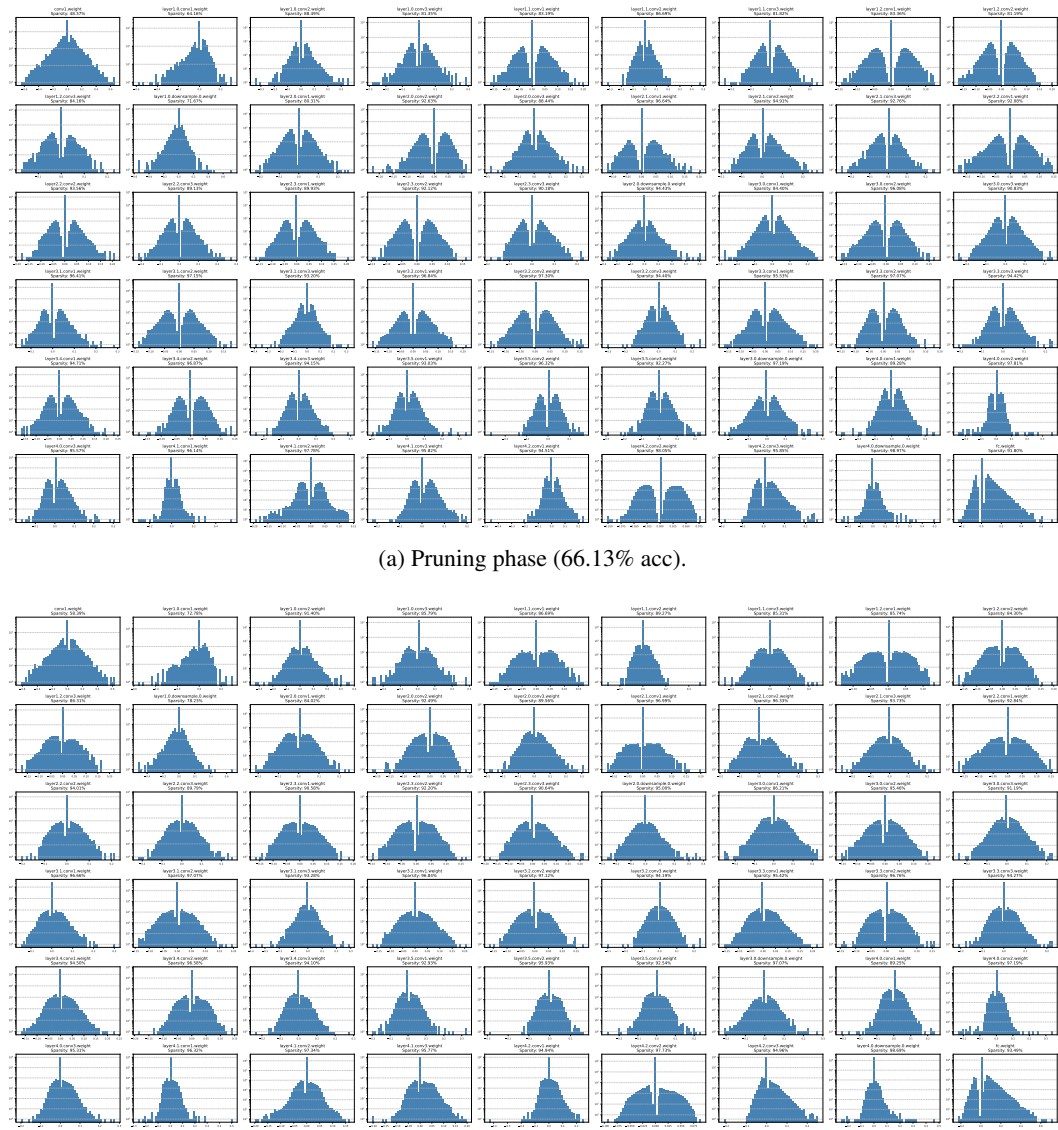

(a) Pruning phase (66.13% acc).

(b) Regrowth phase (72.40% acc).

Figure 12: Weight histograms of ResNet-50 on ImageNet during two different phases at 95.77% sparsity.

# D Complexity analysis

We analyze both the training-time and inference-time compute cost of Hyperflux relative to a standard dense baseline. Let $f_d$ denote the FLOPs required for a forward pass of the dense network, and $f_s$ the (reduced) FLOPs for a forward pass through the sparse weights. We approximate the backward-pass cost for sparse weights as $2f_s$, following common conventions, and account for the dense parameters $t$ with a full backward cost of $f_d$. The reason for this is that all $t$ values are updated, no matter whether their associated weight is pruned or not, thereby requiring the full cost $f_d$. Thus, the total training cost of Hyperflux is

$$\text{FLOPs}_{\text{train}} = f_s \; + \; 2f_s \; + \; f_d = 3f_s + f_d,$$

while the dense baseline requires

$$\text{FLOPs}_{\text{train}}^{\text{dense}} = f_d \; + \; 2f_d = 3f_d.$$

Consequently, the relative training cost is:

$$\frac{3f_s + f_d}{3f_d} = \frac{f_s}{f_d} + \frac{1}{3}.$$

At inference, Hyperflux uses only the sparse weights, yielding:

$$\text{FLOPs}_{\text{test}} = f_s.$$

# E Schedulers implementation

In section 3.4 we discussed briefly about pressure policy $\mathcal{C}$, but we did not provide detailed implementation. Here we provide clear steps that each scheduler follows along with our experimental findings. We begin by defining a mapping from epoch $e$ to a expected decay factor in sparsity at epoch $e$, $0 < d(e) \leq 1$. Our sparsity function $f(e)$ is defined as $f(e) = 100 \cdot \prod_{i=1}^{e} \cdot d(i)$. For example, if $d(1) = 0.9$ and $d(2) = 0.8$, the expected sparsity at epoch 2 will be $f(2) = 100 \cdot 0.9 \cdot 0.8 = 72$. We will use $f(e)$ from now on to refer to our sparsity curve. For the $u$ parameter in our scheduler we find a value of $0.1$ to be suited for vision tasks, while for $\alpha$ we choose $1.5$.

## E.1 Pressure scheduler with trajectory control

This is the first implementation of our scheduler pressure policy $\mathcal{C}$. Its aim is to track a user defined curve at each epoch, by increasing the pressure when sparsity is below the curve (too many parameters) and decreasing pressure when sparsity is above the curve (too few parameters). Concretely, the decisions at each step are taken as in Algorithm 3. This scheduler is able to follow more precisely a specific sparsity trajectory, but its convergence standard deviation to $\mathcal{S}$ might reach 10%-20% of the remaining parameters.

---

**Algorithm 3** Pressure policy $\mathcal{C}$ for trajectory control

---

1: **Initialization:** Sparsity function $f$.
2: **Policy parameteres:** Pruning epochs $E_p$, Final sparsity $\mathcal{S}$, Current sparsity $s_e$, Current epoch $e$,
3: **if** $s_e < f(e)$ **then**
4:     Return true
5: **else**
6:     Return false
7: **end if**

---

## E.2 Pressure scheduler with upper boundary

The second implementation of the pressure policy $\mathcal{C}$ achieves $\mathcal{S}$ within tight boundaries (under 5% standard deviation from expected remaining parameters (e.g. $\mathcal{S} = 98\%$, we expect 2% remaining parameters and the scheduler generally reaches the interval $[1.95, 2.05]\%$), by trading off exact control over sparsity curve. The upper boundary, defined in the same way as the sparsity curve but

used differently, is recalculated at each epoch, essentially creating an ever-tightening sparsity space in which the network's sparsity resides. When the network's sparsity increases (fewer parameters), the upper boundary is recalculated to not allow decreases in sparsity again. The algorithm for $\mathcal{C}$ is presented in Algorithm 4. Since ub is recalculated at each epoch, it does not make sense to access other indexes other than 1. However, we still need to calculate the curve in order to ensure the network trajectory is steered towards $\mathcal{S}$.

---

**Algorithm 4** Pressure policy $\mathcal{C}$ for upper boundary

---

1: **Initialization:** Upper boundary function $ub$.
2: **Policy paramteres:** Pruning epochs $E_p$, Final sparsity $\mathcal{S}$, Current sparsity $s_e$, Current epoch $e$,
3: **Internals:** Sparsity history $sh$.
4: sh.append($s_e$).
5: Recalculate $ub$ such that $ub(E_p - e) = \mathcal{S}$ (will reach $\mathcal{S}$ in the remaining epochs).
6: prev_decrease $\leftarrow \frac{sh(e)}{sh(e-1)}$
7: **if** prev_decrease $< ub(1)$ **then**
8:     Return true
9: **else**
10:     Return false
11: **end if**

---

# F  Training setup and reproducibility

Table 4: Hyperparameter configurations for the pruning and stabilization stages across CIFAR-10, CIFAR-100, and ImageNet-1K with ResNet-50 and VGG19 architectures.

| Dataset | CIFAR-10 | | CIFAR-100 | | ImageNet-1K |
|---|---|---|---|---|---|
| **Network** **Acc (%)** | **ResNet-50** $94.72 \pm 0.05$ | **VGG19** $93.85 \pm 0.06$ | **ResNet-50** $78.32 \pm 0.08$ | **VGG19** $73.44 \pm 0.09$ | **ResNet-50** $77.01$ |
| **Batch size** | 128 | 128 | 128 | 128 | 1024 |
| Total epochs $E_p/E_s$ | 160 100/60 | 160 100/60 | 160 100/60 | 160 100/60 | 100 80/20 |
| $\mathcal{O}_t$ $\mathcal{O}_\omega$ $\mathcal{S}_\omega$ | ADAM SGD Cosine | ADAM SGD Cosine | ADAM SGD Cosine | ADAM SGD Cosine | ADAM SGD Cosine |
| *Pruning* | | | | | |
| $\eta_\omega^s$ $\eta_\omega^e$ $\eta_t$ | 0.1 0.003 0.001 | 0.1 0.003 0.001 | 0.1 0.003 0.001 | 0.1 0.003 0.001 | 0.1 0.003 0.001 |
| *Stabilization* | | | | | |
| $\eta_\omega^i$ $\eta_\omega^f$ $\eta_t$ | 0.001 0.0001 0.001 | 0.001 0.0001 0.001 | 0.001 0.0001 0.001 | 0.001 0.0001 0.001 | 0.001 0.0001 0.001 |
| $\mathcal{S}_t$ $\lambda_t$ | LambdaLR 0.75 | LambdaLR 0.75 | LambdaLR 0.75 | LambdaLR 0.75 | LambdaLR 0.55 |

As summarized in Table 4, our training protocol consists of two stages over a fixed number of epochs: a pruning stage followed immediately by a stabilization (regrowth) stage. In both stages, weights $\omega$ are optimized with by SGD under a cosine annealing scheduler $\mathcal{S}_w$, while presence parameters $t$ are optimized with ADAM. Furthermore, the presence parameters are uniformly initialized in the range 0.2–0.5.

During the pruning stage the learning rate for $\omega$ is decayed from $\eta_\omega^s$ to $\eta_\omega^e$, and $t$ uses a constant $\eta_t$. Without resetting training, we then set the pruning pressure to zero and enter the stabilization stage, where $\eta_\omega$ is further decayed (from its new $\eta_\omega^i$ to $\eta_\omega^f$) and the presence parameters are trained under a LambdaLR scheduler $\mathcal{S}_t$ with decay parameter $\lambda_t$. This two-stage setup, with separate optimizer and learning rate schedules for weights and presence parameters, ensures that both the sparse structure and the remaining weights are allowed to converge to their optimal configurations. Let $\mathcal{O}_w$ be the optimizer for the weights and $\mathcal{O}_t$ the optimizer for the presence parameters. We prune for $E_p$ epochs and then enter a stabilization stage lasting $E_s$ epochs, for a total of $E_p + E_s$ epochs. All experiments use the pressure scheduler presented in Algorithm 3.

Across all experiments, we applied a weight decay of $10^{-4}$, while omitting any weight decay on the batch-normalization layers. Regarding augmentations, on ImageNet we adopt the same pipeline as our baselines: random resize, crop and random horizontal flip for training, and resize plus center crop for validation; on CIFAR-10/100 we apply random crop with padding, random horizontal flip for training, and no augmentations for testing.

