# OpenReview forum: "Hyperflux: Pruning Reveals the Importance of Weights"
_NeurIPS.cc/2025/Conference — Submitted to NeurIPS 2025_

### Official Review · Reviewer_YpEF · 2025-07-01

**Clarity:** 4
**Significance:** 2
**Originality:** 2
**Rating:** 3
**Confidence:** 4

**Summary:**

The paper introduces Hyperflux, a framework for pruning weights in neural networks through the gradient response to weight removal (flux). The framework dynamically prunes and regrows weights based on a 'pressure' term. In addition, the framework allows using a scheduler for controlling sparsity. The method is well-motivated and performs well on CIFAR10/100. On ImageNet, Hyperflux lags slightly behind state-of-the-art methods in accuracy and training efficiency.

**Questions:**

I would appreciate addressing the concerns I've raised in the weaknesses section.

**Ethical Concerns:**

["NO or VERY MINOR ethics concerns only"]

**Final Justification:**

The authors did not resolve my concerns. Hence, I keep my score

**Limitations:**

The authors mention the limitations of the method, which lie mainly in the additional computational cost during training.

**Paper Formatting Concerns:**

I found no formatting issues

**Quality:**

2

**Strengths And Weaknesses:**

Strengths:
1. The paper presents an intuitive idea of estimating weight importance by observing the gradient flow after pruning.
2. The method allows for dynamic pruning where pruned weights can 'regrow' at a later stage.
3. Competitive results on CIFAR10/100 at high sparsity.
4. The proposed scheduling mechanism provides a principled approach to controlling sparsity during training.

Weaknesses:
1. While the idea of 'flux' is well-articulated, it overlaps significantly with other gradient-aware methods. Also, the 'Topology' definition shares many similarities with [1]. The main difference seems to lie in the framing rather than the fundamental technique. This limits the novelty of this work.
2. On ImageNet, Hyperflux results fall slightly behind AC/DC and Spartan, while requiring higher training costs. This is due to the additional parameters and calculations revolving around flux.
3. The evaluation is limited to image classification on CIFAR and ImageNet, and with old network architectures (ResNet will soon be 10 years old). It would strengthen the paper to include more complex tasks (such as semantic segmentation, object detection, and depth prediction) with newer architectures (E.g., ConvNeXt, ViT).


[1] Savarese et al. Winning the Lottery with Continuous Sparsification. NeurIPS 2020.

---

> ### Author Rebuttal · Authors · 2025-07-29
>
> We thank the reviewer for going over the paper and identifying both valuable weaknesses and strengths. We are addressing questions and weaknesses below.
>
> **“While the idea of 'flux' is well-articulated, it overlaps significantly with other gradient-aware methods. Also, the 'Topology' definition shares many similarities with [1]. The main difference seems to lie in the framing rather than the fundamental technique. This limits the novelty of this work.”**
>
> The core novelty of Hyperflux lies in its conceptual framework, while the mechanism itself is also unique.
>
> Hyperflux builds a conceptual framework around the pruning process. We analyze the gradients interactions and properties in section 3.2 from which we derive our flux/pressure framework. We test its predictive power by postulating and validating its properties in section 3.3. We arrive at non-trivial properties, such as the Neural Pruning law relationship, which hints at connections between pressure-sparsity and Neural scaling laws.
>
> Furthermore, even if the idea of gradient-aware regrowth is not new, the system as a whole is. While gradient-aware methods such as GraNet, RigL, etc, attempt to estimate weight importance, Hyperflux takes another approach, by observing how pruning affects performance. Furthermore, many methods have a fixed explicitly imposed pruning-regrowth loop, while in Hyperflows this loop emerges as a natural side effect of the interactions between gradients, leading to unique behavior for each weight as seen in Appendix C2.
>
> **“On ImageNet, Hyperflux results fall slightly behind AC/DC and Spartan, while requiring higher training costs. This is due to the additional parameters and calculations revolving around flux.”**
>
> We acknowledge the higher training costs of our method and consider it a fair trade-off of our method. This is due to the design design philosophy of Hyperflux, which aims for empirically strong results and understanding about the pruning process as opposed to only empirical performance with as few FLOPs as possible.
>
> Hyperflux is not falling behind Spartan as shown in Table 3 in Section 4.2. We ran all the methods using the same training recipe, as detailed in lines 288-291, including Spartan.
>
> **“The evaluation is limited to image classification on CIFAR and ImageNet, and with old network architectures (ResNet will soon be 10 years old). It would strengthen the paper to include more complex tasks (such as semantic segmentation, object detection, and depth prediction) with newer architectures (E.g., ConvNeXt, ViT).”**
>
> The benchmarks used are standard in the pruning field and provide a strong evaluation. To further strengthen the claims made in the paper, we will add additional experiments involving Vision transformers in the final version of the manuscript.

---

> > ### Comment · Reviewer_YpEF · 2025-08-04
> >
> > I thank the authors for the response.
> >
> > The analysis of the method's properties is appreciated. However, I do not see how those properties make the method more compelling, or how they can be generalized to other methods.
> >
> > I agree with the authors that CIFAR and ImageNet classification are, unfortunately, still the standard in the field. It is known in practice that dense predictors suffer more from network compression (including weight quantization and pruning). Hence, I believe that strong results in these areas can strengthen the paper's claims for "empirically strong results", which are currently not evident.

---

### Official Review · Reviewer_CGEM · 2025-07-02

**Clarity:** 3
**Significance:** 2
**Originality:** 4
**Rating:** 2
**Confidence:** 5

**Summary:**

This paper presents Hyperflux, a L0 pruning approach that estimates each weight’s importance through its flux, the gradient’s response to the weight’s removal. A global pressure term continuously drives all weights toward pruning, with those critical for accuracy being automatically regrown based on their flux. The paper's evaluation state-of-the-art results with ResNet-50 and VGG-19 on CIFAR-10 and CIFAR-100.

**Questions:**

- Why does HyperFlux perform worse than AC/DC?
- How long does HyperFlux take to prune a model like ResNet-50?

**Ethical Concerns:**

["NO or VERY MINOR ethics concerns only"]

**Final Justification:**

Given the large body of existing works, is still unclear what is fundamentally new in the proposed work that could help it outperform SOTA or could be generalized to explain pruning beyond the proposed specific method.

**Limitations:**

Can the proposed solution generate real societal impact to widely deployed real-world applications and systems?

**Quality:**

2

**Strengths And Weaknesses:**

Strengths:
+ The idea of pruning model weights based on flux is original and interesting.
+ The paper is able to explain why the proposed method works and experimentally validate the claims.
+ The paper is well written.

Weaknesses:
- The considered architectures include only ResNet-50 and VGG-19 and are not diverse enough. These models are also both quite outdated.
- The improvement over SOTA is small. Given the large number of related pruning works with similar performance, this paper appears to be just yet another pruning work and its significance is questionable.
- All the considered baselines were published before 2022.
- The proposed method in fact performs worse than the related work AC/DC.
- As a unstructured pruning method, the resulting models do not run well on commodity hardware.

---

> ### Author Rebuttal · Authors · 2025-07-29
>
> We appreciate the time and effort invested into the constructive feedback on the paper provided by the reviewer. We clarify the questions and address the weaknesses below.
>
> **“The considered architectures include only ResNet-50 and VGG-19 and are not diverse enough. These models are also both quite outdated."**
>
> While the architectures are not new, they are still standard benchmarks used in recent pruning papers, such as GraNet, AC/DC, Feather, CAP, etc. Thus, there was a wide range of papers already using them which we can compare Hyperflux. To further strengthen the experiments, we will add vision transformers in the final version of the manuscript.
>
> **“The improvement over SOTA is small. Given the large number of related pruning works with similar performance, this paper appears to be just yet another pruning work and its significance is questionable.”**
>
> Large improvements in accuracy were not the purpose of the paper. Instead, our objective was to advance conceptual understanding of pruning while maintaining strong performance, as mentioned on lines 31-32. We consider a work such as ours which achieves comparable to SOTA or SOTA results to be empirically strong.
>
> What differentiates Hyperflux from other pruning works is the conceptual framework we built around our method which has significant predictive power. After the in-depth analysis of Hyperflux mechanisms performed in Section 3.2, we postulate several properties that emerge from our flux/pressure framework, which we validate empirically afterwards. For example, eq. (6) and the definition of flux directly lead to property 1 (network convergence). These properties are not trivial and are enabled precisely by our conceptual framework, demonstrating concrete value. The power-law relationship between sparsity and pressure (Property 2) is especially interesting, since it hints at deeper connections to neural scaling laws.
>
> We consider that our framework advances understanding of pruning and provides explainability, showing that pruning should not be a black box as it is often regarded. This differentiates us from papers in the literature.
>
> **“All the considered baselines were published before 2022.”**
>
> We acknowledge this, however, if one were to look at the SOTA results on ImageNet pruning, we can see that all but one benchmark are published before 2022 (https://hyper.ai/en/sota/tasks/network-pruning/benchmark/network-pruning-on-imagenet-resnet-50-90). For the benchmark published in 2023, Feather, we were not able to find reliable code to run the method. This was also the case with CIFAR10/100 on VGG19 and ResNet50 benchmarks.
>
> **“The proposed method in fact performs worse than the related work AC/DC.”
> “Why does HyperFlux perform worse than AC/DC?”**
>
> The main reason for this under-performance is the design philosophy of Hyperflux and AC/DC. In Hyperflux, we aim at creating a dynamic pruning method, around which we build a conceptual framework of understanding about the pruning process, even at a slight expense in empirical results (0.7% difference in accuracy).
>
> **“How long does HyperFlux take to prune a model like ResNet-50?”**
>
> Hyperflux uses about 0.5 of the normal training FLOPs for pruning a model such as Resnet-50. FLOPs numbers can be found in table 3 in section 4.2 for ImageNet. The reasoning behind these numbers is in the appendix in section D. For wall clock number in the case of ResNet-50 ImageNet experiment, the mean running time on our 3RTX 4090 is about 16 hours.
>
> **“As a unstructured pruning method, the resulting models do not run well on commodity hardware.”**
>
> We believe that the development of unstructured pruning methods will advance the general knowledge in this area. These contributions will lead to stronger structured pruning methods.
>
> **“Can the proposed solution generate real societal impact to widely deployed real-world applications and systems?”**
>
> Yes, network pruning reduces inference latency, aiding in the creation of real-time systems, reducing energy and cooling costs and thereby reducing pollution.

---

> > ### Comment · Reviewer_CGEM · 2025-08-03
> >
> > "What differentiates Hyperflux from other pruning works is the conceptual framework we built around our method which has significant predictive power."
> >
> > Can you provide some use cases of this "predictive power"?

---

> > > ### Comment · Reviewer_CGEM · 2025-08-07
> > >
> > > It is still unclear to me what is fundamentally new in the proposed work that could help it outperform SOTA or could be generalized to explain pruning beyond the proposed specific method. Therefore, I maintain my original rating.

---

> ### Author Response · Authors · 2025-08-03
>
> The predictive power in Hyperflux is used to discover the properties in section 3.3. Each of these properties has their individual use cases.
>
> For example, Property 2 (lines 224-238) motivates the non-linearity in our scheduler’s update rule (lines 262-267). Property 2 also provides a way to predict the exact pressure needed to produce a certain desired sparsity.

---

### Official Review · Reviewer_po7G · 2025-07-02

**Clarity:** 2
**Significance:** 3
**Originality:** 3
**Rating:** 3
**Confidence:** 4

**Summary:**

This paper introduces Hyperflux, which prunes a network by adding a single pressure term γ that keeps pushing every weight toward zero.
When a weight is set to zero, the method looks at the new gradient—called flux—to see how badly the loss wants that weight back; big flux means the weight will grow again. Because pruning (pressure) and regrowth (flux) run together at every step, the model keeps only the weights it truly needs. A simple rule links γ to the target sparsity, so a built-in γ-scheduler can hit any desired sparsity without manual tuning. On CIFAR-10/100 and ImageNet-1k, this approach matches or beats RigL, GraNet, and AC/DC while staying up to 96 % sparse and using just 6–15% of the dense FLOPs.

**Questions:**

My main concern is that the novelty of the proposed method is not very clear to me (see Cons #1 above). If the authors can clarify this point (as well as other cons listed above), I would be willing to raise my score.

**Ethical Concerns:**

["NO or VERY MINOR ethics concerns only"]

**Final Justification:**

I have read the comments of other reviewers. I appreciate the author’s detailed response.

**Limitations:**

Yes.

**Paper Formatting Concerns:**

No paper formatting concerns.

**Quality:**

3

**Strengths And Weaknesses:**

Pros
1. I found the flux–pressure idea very interesting. By defining “flux” as the gradient that appears when a weight is gone, the method ties importance directly to loss change, giving a more interpretable alternative to magnitude, Taylor expansion, or Hessian heuristics.

2. The experiment results look good. For example, on CIFAR-10/100 and ImageNet-1k, the method matches or surpasses strong baselines such as RigL, GraNet and AC/DC at sparsity levels up to 96 %, while cutting inference FLOPs to as low as 6–15 % of the dense model.

3. The comprehensive ablation study further validate the effectiveness of the proposed method.


Cons
1. My main concern is that the novelty of the proposed method might be relatively limited.
Flux is essentially a first-order Taylor term already explored in earlier Taylor-based pruning work [1], the author also shows that the proposed method has a substantial link to the Taylor expansion based existing works in Appendix. The prune-regrow loop follows the Dynamic Sparse Training (DST) line started by SET [2], RigL [3] and GraNet [4]. The paper mainly combines these two ideas (though with some incremental but solid improvement) rather than introducing a new paradigm.

3. I think the training cost of the proposed method is quite high.
Training still requires about 0.5–0.6× the dense FLOPs on ImageNet, which is higher than GraNet and similar DST baselines; the paper gives no wall-clock or memory numbers.

4. All results focus on image classification, so the claimed generality remains unknown.


[1] Molchanov, Pavlo, Arun Mallya, Stephen Tyree, Iuri Frosio, and Jan Kautz. "Importance estimation for neural network pruning." In Proceedings of the IEEE/CVF conference on computer vision and pattern recognition, pp. 11264-11272. 2019.

[2] Mocanu, Decebal Constantin, Elena Mocanu, Peter Stone, Phuong H. Nguyen, Madeleine Gibescu, and Antonio Liotta. "Scalable training of artificial neural networks with adaptive sparse connectivity inspired by network science." Nature communications 9, no. 1 (2018): 2383.

[3] Evci, Utku, Trevor Gale, Jacob Menick, Pablo Samuel Castro, and Erich Elsen. "Rigging the lottery: Making all tickets winners." In International conference on machine learning, pp. 2943-2952. PMLR, 2020.

[4] Liu, Shiwei, Tianlong Chen, Xiaohan Chen, Zahra Atashgahi, Lu Yin, Huanyu Kou, Li Shen, Mykola Pechenizkiy, Zhangyang Wang, and Decebal Constantin Mocanu. "Sparse training via boosting pruning plasticity with neuroregeneration." Advances in Neural Information Processing Systems 34 (2021): 9908-9922.

---

> ### Author Rebuttal · Authors · 2025-07-29
>
> We thank the reviewer for the time taken to review Hyperflux and connecting our work to the broader literature of Taylor-based methods and Dynamic Sparse Training (DST). We address the novelty concern as well as the difference to existing methods and other concerns below.
>
> **“My main concern is that the novelty of the proposed method is not very clear to me (see Cons #1 above). If the authors can clarify this point (as well as other cons listed above), I would be willing to raise my score.”**
>
> The core novelty of Hyperflux lies in the conceptual framework we developed around our method and its predictive power. The framework is built upon analyzing the gradients and their interactions in Section 3.2 (e.g. regrowth occurs only if flux exceeds pressure and prune-regrowth appear as a consequence), and its predictive relevance is tested by postulating properties and then verifying them empirically.
>
> For example, as a direct consequence of eq(6) in Section 3.2 we postulate that sparsity convergence should occur if the network is left to train for enough time, which is what we observe after hundreds to thousands of epochs in Figure 3a. Furthermore, we find a generalized power-law equation between pressure and sparsity, hinting at connections to the broader field of Neural Scaling Laws.
>
> We believe that these properties are non-trivial and their discovery is enabled by our framework.
>
> **“Flux is essentially a first-order Taylor term already explored in earlier Taylor-based pruning work [1]”**
>
> Prior Taylor-based work [1] use the Taylor series to approximate the effect of removing the weight, while the weight is present. In our method, we first remove the weight and then approximate the effect of adding the weight back to the network. The resulting flux (when the weight is removed) is not an approximation of importance but a direct measure of the response of the loss to the absence of that weight. This moves beyond estimation to a causal measurement.
>
> **“The prune-regrow loop follows the Dynamic Sparse Training (DST) line started by SET [2], RigL [3] and GraNet [4].”**
>
> While superficially similar to Dynamic Sparse Training, our prune-regrow mechanism is fundamentally different from the explicit, global mechanism of DST methods (SET [2], RigL [3] or GraNet [4]). In those methods, a fixed budget of weights is pruned and another is regrown in discrete steps. In Hyperflux, there is no explicit loop. Instead, prune-regrow cycles are emergent, continuous, and per-weight, arising from the local competition between the global pressure and each weight's individual flux. As shown in Appendix C2 (Fig. 10), this leads to highly diverse behavior in each weight's prune and regrowth patterns.
>
> **“I think the training cost of the proposed method is quite high. Training still requires about 0.5–0.6× the dense FLOPs on ImageNet, which is higher than GraNet and similar DST baselines; the paper gives no wall-clock or memory numbers.”**
>
> We acknowledge training the cost of Hyperflux and consider it a trade-off of our method. We considered FLOPs to be an objective metric, as they are not tied to the physical hardware used to run the neural networks. For wall clock number in the case of ResNet-50 ImageNet experiment, the mean running time on our 3RTX 4090 is about 16 hours.
>
>
> **“All results focus on image classification, so the claimed generality remains unknown”**
>
> We choose image classification as it is a common benchmark in pruning literature, and we intend to extend our experiments to other benchmarks in future works.

---

> > ### Comment · Reviewer_po7G · 2025-08-07
> >
> > I appreciate the author’s detailed response. Most of my concerns have been addressed well. I have raised my score. I would be happy to see this work extended in the future. For example, if it can be applied to other tasks such as detection, its generality would be further improved.

---

### Official Review · Reviewer_tjNh · 2025-07-06

**Clarity:** 3
**Significance:** 3
**Originality:** 4
**Rating:** 5
**Confidence:** 3

**Summary:**

The paper proposes a pruning algorithm called hyperflux. The method works by adding an learnable 'importance' parameter $t_i$ for each weight $w_i$ which is activated by a step-function and multiplied in its corresponding weight as $w_i \times H(t_i)$. During backpropagation the gradient of $H$ is taken to be identity (which can be considered as a kind of straight-through estimation). The authors derive the gradients and use it to describe the behavior of the algorithm. A term (called 'pressure') proportional to $\sum t_i$ is added to the loss function to drive weights towards pruning. They demonstrate state-of-the-art results with ResNet-50 and VGG-19 on CIFAR-10 and CIFAR-100.

**Questions:**

* Why do the 90% and 95% pruned networks with hyperflux consistently beat the original dense network on CIFAR10 and CIFAR100?
* The three row groups in Table 3 aren't labeled. What are they?
* Are you reporting test accuracy or training accuracy?

**Ethical Concerns:**

["NO or VERY MINOR ethics concerns only"]

**Final Justification:**

I like the creative algorithm that the authors propose. However, I also see the issues raised by other reviewers: yet another pruning method, worse than AC/DC, etc. I can agree that having one creative idea should perhaps not put a paper above the bar of acceptance for neurips so I am OK if the decision ends up being reject for this paper. Perhaps another venue is more suitable for such explorative work that is not yet compelling enough.

**Limitations:**

Yes.

**Paper Formatting Concerns:**

None.

**Quality:**

3

**Strengths And Weaknesses:**

## Strengths

* The paper is clearly written and the mechanism with which pruning is achieved is clearly described.
* I find the method to be very novel and creative (but note that I'm not very familiar with the area of pruning). For example, it is interesting that they have two variables for each weight. Commonly the magnitude of a weight is assumed to be its importance but, here, the two are separated. Novel ideas can potentially be very impactful.
* Strong empirical performance: state-of-the-art on CIFAR10 and CIFAR100 and competitive with state-of-the-art on ImageNet.

## Weaknesses

* I wouldn't say that the method is "conceptually grounded." Deriving gradients isn't a theoretical justification for the method. But it's fine to leave a more theoretical understanding for future works.

---

> ### Author Rebuttal · Authors · 2025-07-29
>
> We thank the reviewer for the valuable and positive feedback on Hyperflux. Below we address the questions and weaknesses identified.
>
> **“I wouldn't say that the method is "conceptually grounded." Deriving gradients isn't a theoretical justification for the method."**
>
> The conceptual grounding in our work goes beyond the simple mathematical derivations themselves. In the second half of section 3.2, we analyze the interactions and behavior of the gradients, relating them to flux and pressure. This framework exhibits predictive power, being able to postulate non-obvious properties about pruning dynamics which are then validated in section 3.3. This represents an improvement over other pruning methods, which do not analyze their pruning mechanisms in detail nor build a conceptual framework around their procedures.
>
> **“Why do the 90% and 95% pruned networks with hyperflux consistently beat the original dense network on CIFAR10 and CIFAR100?”**
>
> As noted in lines 194 and 288–291, we begin pruning by initializing the network with pretrained weights, as our method relies on parameters that encode meaningful information. Then our method naturally examines these weights across multiple topologies, identifying those that yield improved generalization. In doing so, the 90 % and 95 % sparse versions consistently discover weight configurations that outperform the original dense network on both CIFAR‑10 and CIFAR‑100.
>
> **“The three row groups in Table 3 aren't labeled. What are they?”**
>
> The three row groups represent comparisons between several state-of-the-art methods and Hyperflux on ResNet-50 ImageNet, at the same sparsity. The first group evaluates all methods at 90% sparsity, the second group at 95% sparsity and the third group evaluates the methods at 96.5% sparsity. The “s” column in the table refers to sparsity, this will be explicitly mentioned in the final paper.
>
> **“Are you reporting test accuracy or training accuracy?”**
>
> We are reporting test accuracy, as it is standard in the literature. We will ensure this is explicitly mentioned in the final manuscript.

---

> > ### Comment · Reviewer_tjNh · 2025-08-06
> >
> > I thank the authors for responding to my questions.

---

### Decision · Program_Chairs · 2025-09-17

**Decision:**

Reject

**Comment:**

In this work, the authors propose a strategy to perform unstructured pruning leveraging the knowledge of *flux*, namely the effect of the parameter removal estimated through the gradient. This is evaluated on small-scale datasets, and ImageNet.

During rebuttal, one reviewer supported full acceptance, mainly praising the creativity of the proposed method. The other three reviewers were aligned to reject the paper, for multiple reasons ranging from the fact that the true contribution in terms of pure results is unclear to the practical utility of the work, and even remaining unsure of the concrete merits of the work.

Related to the novelty, there are some works that are practically regularizing during training to enforce sparsity. While the AC agrees that there could be some interesting elements arising from a different perspective analysis, technically, the ideas are very close to some of the referenced literature (like those indicated in the paragraph "Dynamic pruning", which should be better detailed to discriminate the elements of novelty). Other relevant examples of gradient-based methods to discuss and/or to take inspiration for an update of the proposed experiment section can be found as an example here [A-C].

Another weakness to consider is how to set a-priori the new hyperparameters, namely the scheduling parameters steering $\gamma$. Besides, the gains in FLOPs terms seem to be only theoretical. Finally, the experimental setup is unanimously considered by the reviewers as outdated, although necessary. One suggestion is to move the method to LLMs, as for example done in [A, C] and other recent works. Indeed, the fact that Hyperflux is (marginally) worse than other approaches when scaling to ImageNet (while being the best in most of the smaller scale experiments), raises questions about the validity on even larger models.

The authors are encouraged to continue working on the paper, focusing on the points raised during the review phase.


[A] Gao, Yuan, et al. "Bypass back-propagation: Optimization-based structural pruning for large language models via policy gradient." arXiv preprint arXiv:2406.10576 (2024).

[B] Tartaglione, Enzo, et al. "Loss-based sensitivity regularization: towards deep sparse neural networks." Neural Networks 146 (2022): 230-237.

[C] Fontana, Federico, et al. "Distilled gradual pruning with pruned fine-tuning." IEEE Transactions on Artificial Intelligence 5.8 (2024): 4269-4279.